# A Convolutional Neural Network-Based Method for Corn Stand Counting in the Field

**DOI:** 10.3390/s21020507

**Published:** 2021-01-13

**Authors:** Le Wang, Lirong Xiang, Lie Tang, Huanyu Jiang

**Affiliations:** 1College of Biosystems Engineering and Food Science, Zhejiang University, 866 Yuhangtang Road, Hangzhou 310058, China; wangle5994@zju.edu.cn; 2Department of Agricultural and Biosystems Engineering, Iowa State University, Ames, IA 50011, USA; xiang@iastate.edu (L.X.); lietang@iastate.edu (L.T.)

**Keywords:** deep learning, YoloV3, video tracking, corn stand counting

## Abstract

Accurate corn stand count in the field at early season is of great interest to corn breeders and plant geneticists. However, the commonly used manual counting method is time consuming, laborious, and prone to error. Nowadays, unmanned aerial vehicles (UAV) tend to be a popular base for plant-image-collecting platforms. However, detecting corn stands in the field is a challenging task, primarily because of camera motion, leaf fluttering caused by wind, shadows of plants caused by direct sunlight, and the complex soil background. As for the UAV system, there are mainly two limitations for early seedling detection and counting. First, flying height cannot ensure a high resolution for small objects. It is especially difficult to detect early corn seedlings at around one week after planting, because the plants are small and difficult to differentiate from the background. Second, the battery life and payload of UAV systems cannot support long-duration online counting work. In this research project, we developed an automated, robust, and high-throughput method for corn stand counting based on color images extracted from video clips. A pipeline developed based on the YoloV3 network and Kalman filter was used to count corn seedlings online. The results demonstrate that our method is accurate and reliable for stand counting, achieving an accuracy of over 98% at growth stages V2 and V3 (vegetative stages with two and three visible collars) with an average frame rate of 47 frames per second (FPS). This pipeline can also be mounted easily on manned cart, tractor, or field robotic systems for online corn counting.

## 1. Introduction

Stand counting is an important task in breeding and crop management, providing farmers and breeders with information about the germination rate and population density of crops. In the breeding stage, the corn stand count can provide the seed germination rate for breeders and geneticists. As for the planting stage, stand counting can provide the plant population density to farmers, which is an important trait for crop yield and quality [1,2,3,4,5,6,7]. This information can also help farmers with some essential management practices, such as determining the necessity of replanting [8].

The traditional method for stand counting usually samples the corn population manually with quadrats [9], which is time consuming, laborious, and cost intensive. In addition, manual stand counts are not feasible for large fields and are prone to human error. Recent advances in machine vision provide automatic methods for farmers and breeders to count plant seedlings and determine yield-related traits such as germination rate and plant population density. One popular method has been to use a fixed camera on an unmanned aerial vehicle (UAV) to capture digital images of plants, then process the images to obtain the plant stand count offline [10,11,12,13,14].

UAV-based stand counting usually takes images with camera heights varying from 10 m to 50 m [11,12,13], with some even over 100 m [15,16]. Thus, it is challenging to acquire a high-resolution image of every single plant, especially for the small plants that are in their early stages. As a result, these small objects cannot be detected accurately. Apart from UAVs, there are many ways to collect seedling images, such as moving-cart-based systems [15], tractor-based systems [17,18], and automatic field robots [19], which can collect images and information accurately and efficiently [20]. Besides RGB-camera-based methods, there are also some other sensors that can be used for object counting in the field, such as Lidar sensors [21,22]. 

Detecting plants accurately from images is the foundation of image-based stand counting. Different methods and their counting accuracies are summarized in Table 1. UAV-based stand counting usually requires a stitching and alignment operation. Some commercial software such as Agisoft Photoscan (Agisoft LLC, St. Petersburg, Russia) and Pixel4D (Pix4D S.A., Lausanne, Switzerland), as well as some algorithms, have been developed [11] to align UAV-collected images automatically. Then objects need to be identified from the images. In traditional image processing, different plant features have been extracted or designed to identify objects, like edge [23], color [24] and shape [12]. Some are also equipped with a hyperspectral sensor, and identify objects through NDVI index [11].

Since the significant evolution of computation in 1999, graphics processing units (GPUs) have been developed for image processing and a revolution has taken place in the application of deep learning. State-of-the-art networks include the faster region-based convolutional network (Faster R-CNN) [25], the mask region-based convolutional network (Mask-RCNN) [26], and You Only Look Once (Yolo) [27,28,29]. It is a promising method to develop algorithms for agricultural object detection and counting based on deep convolutional neural networks (CNNs) [13,17,30]. In particular, Faster R-CNN [25] and Yolo [28] are representative two-stage and one-stage detection networks, respectively. Each of them has made significant breakthroughs in the detection of different objects. Faster R-CNN features high accuracy, while Yolo features a high processing speed. The two networks have been widely applied in agricultural object detection. For example, DeepSeedling integrated Faster R-CNN as the target detection network, then developed a framework to track and count cotton seedlings in the videos [17]. In addition, Faster R-CNN was used as the detection network for detecting other agricultural objects such as plant leaves [31], maize seedlings [32], kiwi fruit [33], grapes [34], and apples [35]. However, most Faster-R-CNN-based approaches are offline. Real-time detection for similar objects usually implements the Yolo network [36,37,38,39,40], which can reach an F1 score of 0.817 for apple detection (with YoloV3) [39], and 0.944 for mango detection (with YoloV3-tiny) [36]. For this study, we chose the Yolo model to satisfy the online counting requirement.

The above image-based counting studies have achieved satisfactory counting accuracies (Table 1). For example, Rawla et al. (2018) achieved corn stand counting with an accuracy of over 99%, using a MicaSense RedEdge Multispectral UAV Camera with a 6 mm lens, mounted on a Hexacopter UAV. The images were processed manually through an ImageJ-based plugin [41]. Cameras equipped with expensive hyperspectral sensors can also achieve 98% count accuracy [11]. These methods are costly, however, both in money and time.

In summary Table 1, the most commonly used counting system is UAV-based, but no real-time counting system has really been developed. The UAV-based counting systems have two major disadvantages. The first is a need for reliable hardware. A stable UAV is the basic piece of equipment, and a high-resolution camera is also needed to collect detailed information on plants from a long distance. Expensive sensors are added to improve the count accuracy. The second disadvantage is a reliance on trained experts. A trained pilot is needed to operate the UAV expertly, which is essential to the success of image acquisition and plant detection. In some methods, the image preprocessing stage also requires human intervention [8,11,12,23,42]. Apart from this, battery life and payload are also limitations if considering long-duration work.

Our objective for the study was to develop a machine learning-based image processing pipeline for real-time early-season corn stand counting, which could be integrated into an unmanned robotic system for plant growth monitoring and phenotyping. This proposed corn stand counting pipeline was expected to be an accurate, efficient, and reliable solution for automatically counting early-season corn plants. No extra sensor was needed, only a camera. No professional operator was needed. The system could be easily mounted on manned cart, tractor, or field robotic systems for counting similar plants, like sorghum.

## 2. Materials and Methods

### 2.1. Experimental Setting

One week after plant emergence, video sequences were collected in corn plots planted on 2 August and 30 August 2019 at the Iowa State University Agronomy and Agricultural Engineering Research Center in Boone, Iowa (Figure 1a). Considering the uneven field, we chose a Garmin VIRB Ultra 30 action camera, with 3-axis image stabilization that captures smooth and steady video. The camera was mounted on a cart at 0.5 m above the ground, at the top of the corn seedlings. The camera’s field of view (FOV) was 118° × 69.2°. The video was captured at 30 frames per second (FPS) with a resolution of 3840 × 2160 pixels. The cart was manually pushed between crop rows at about 1 m/s, with the camera looking perpendicularly down over the crop row (Figure 1b). There were 12 rows of corn plants in total.

According to the identification system, corn development can be divided into vegetative (V) and reproductive (R) stages [44]. The V stages are designated numerically as V(n), where (n) represents the number of leaves with visible collars. We collected videos for plants from stages V1 to V4, which are the vegetative growth stages of corn plants when the first, second, third, and fourth leaf collars are visible. The row spacing of corn plants was 0.76 m (30 inches). The number of plants in each row was counted manually as ground-truth data.

### 2.2. Image Preparation

For training our detector model, a total of 7.2 GB of video was collected, and one image per 10 frames was extracted. There was nearly 1.5 GB per row, each of which was 145 feet. Labelme software [45] was used to draw a bounding box to annotate corn plant seedlings, and 864 images with corn plants at stages V1–V4 were labeled. The labeled images were split into train and validation, with a proportion of 722:142. In order to fit the detector model and improve training efficiency, the images were cropped and resized to 1024 × 1024 pixels. The model was trained on a desktop workstation with an Intel@ Xeon^®^ CPU E5-2637 v3@3.5GHz ×16, and an NVIDIA TITAN GPU (Pascal) with 31.3GiB memory. For evaluation, a total of 54.9 GB of video was used, varying from the V1 to V4 stages.

### 2.3. Corn Plant Stand Counting Pipeline

The tracking-by-detection framework is a reliable and robust tracking method for the problem of multiple object tracking (MOT), where objects are detected in each frame and represented as bounding boxes. In this study, we developed a deep-learning-based automated pipeline to detect and count corn plant seedlings (Figure 2a). Firstly, the detection *det*_0_ in the first frame f_0_ generated by YoloV3 is initialized as tracker *trk*_0_. In the second frame, a Kalman filter [46] is used for updating. According to the initialized bounding boxes *trks*_0_, the position of objects in the second frame is predicted and denoted as *trks_predict_*. Then the tracked bounding boxes are associated with bounding boxes *det*_1_ detected in the new frame with the Hungarian algorithm. At this stage, a minimum intersection-over-union (IoU) is added to double-check the association between *trk_predict_* and *det_1_*. If the IoU over associated trks_predict_ and *det_1_* is smaller than the pre-defined IoU threshold, the trks_predict_ will be rejected. After association processing, there are three classes of results: trackers associated with new detection, unmatched detections, and unmatched tracking. Unmatched detections will be assigned new tracking IDs. As for the unmatched tracking, each ID owns its validity, named as *max_age*, which will be abandoned if it is not updated in the next few frames (total *max_age* number of frames). Then, the updated tracking IDs will be passed to the next epoch (Figure 2a). Finally, the counting stage is reached. We defined a finish line (the yellow line) at the bottom of the image (Figure 2b). When the trace of two tracking boxes’ center points, from frame *i-1* to frame *i*, crossed the finish line, we counted 1, shown in Figure 2b as “count”. 

#### 2.3.1. YoloV3 for Corn Plant Seedling Detection

In this study, both YoloV3 and YoloV3-tiny were trained to find the optimal model that works effectively with corn plants at various growth stages. Both of the models used Darknet [47] as the backbone network, which originally had 53 network layers trained on ImageNet [48]. Based on YoloV3, some feature layers were removed, and only two independent prediction branches have been retained for YoloV3-tiny. YoloV3-tiny has 22 layers in total, and the simplified structure results in a tiny file of only 34.7 MB, with an input resolution of 1024 × 1024 in this study. YoloV3 stacked 53 more layers onto the backbone, producing a 106-layer fully convolutional underlying architecture. In YoloV3, detection is carried out by applying 1 × 1 detection kernels on feature maps of three different sizes at three different stages in the network. The shape of the detection kernel is 1 × 1 × (B × (5 + C)), where B is the number of bounding boxes that a cell on the feature map can predict, “5” includes the four bounding box attributes and one object confidence score, and C is the number of classes. Taking the YoloV3 model trained on our dataset as an example (Figure 3), B = 3 and C = 1, so the kernel size is 1 × 1 × 18. The feature map produced by this kernel has an identical height and width to the previous feature map [28]. YoloV3 makes predictions at three scales, which are given precisely by downsampling the dimensions of the input image by 32, 16, and 8. On the other hand, YoloV3-tiny only keeps two feature maps for prediction, 13 × 13 and 26 × 26.

#### 2.3.2. Corn Plant Seedling Tracking and Counting

Once an object has been detected, it can be tracked along its path with a Kalman filter. The Kalman filter technique is a recursive solution to estimate the state of a linear system [46]. There are two main parts to this task: time update and measurement update. The time update equations are responsible for projecting the current state forward (current frame objects, detected bounding boxes) and estimating error covariance, to obtain an a priori estimate for the next frame. The measurement update equations are used to associate this estimation with the system feedback. The bounding boxes detected by YoloV3-tiny are annotated as $Z_{k}$, where the subscript k indicates the discrete time. The object is to estimate a posteriori $\hat{X_{k}},$ which is a linear combination of the priori estimate and the new measurement $Z_{k}$ [49,50].

From the network detection, we get the measurement $Z={\left[u,v,s,r\right]}^{T}$. In order to predict the bounding box in the next frame, we use the following estimation model:(1)$$X={\left[u,v,s,r,\dot{u},\dot{v},\dot{s,}\dot{r}\right]}^{T}$$
where *u* and *v* represent the horizontal and vertical pixel location of the target center, while *s* represents the scale of area of the target’s bounding box, and *r* is the aspect ratio of bounding box width and height. The video was taken by a camera on a manually pushed cart moving at a roughly constant speed, so the inter-frame displacement of each object was approximated as a constant velocity in the model. Unlike the tracking state model for normal bounding boxes [17,51], which takes the aspect ratio *r* as a constant, *r* was considered as a variable that changed over time with noise in this study. The variations in *r* can be attributed to (1) the bounding box of the same plant changing with the camera view angles, especially for the plants in V3 and V4, and (2) the aspect ratio of the same object changing because of leaf movements caused by wind, especially for stage V4. As shown in Figure 4, one individual plant was detected in consecutive video frames with different aspect ratios of the bounding box along the movement axis.

The usual way to count tracking objects is by counting valid tracking IDs, but the detection accuracy of YoloV3-tiny is not sufficient to satisfy the counting requirement. If the detecting network misses an object in several frames and redetects it, the old tracking ID will be abandoned and a new one will be assigned. Apart from this, the Kalman tracker is not so robust when plants appear from the field margin to the center, where the detector can detect the plant but the tracker treats it as a new object. Thus, the results of counting plants by tracking IDs will be much larger than the ground truth. Therefore, we defined a finish line CD (the yellow line in Figure 5) to count the objects. The center point A of a Tracker *i* was connected with the center B of the next Tracker *I + 1*. The intersection of AB and CD represents a valid tracking ID passing the finish line.

## 3. Results

### 3.1. Detection

Experiments showed that both YoloV3 and YoloV3-tiny had high mAP (mean average precision) with corn plants at growth stages V1–V4 (Figure 6). Figure 6 shows detection results with mAP@IoU 0.5 at different growth stages. The mAP@IoU 0.5 means that average precision at the IoU threshold is set as 0.5, which means when IoU for the prediction is over 0.5, we classify the prediction as True Positive. We compared detection performance with plants at stages V1–V4, and the detection results varied in different growth stages. When overlapping occurred, the detection accuracy dropped sharply.

The YoloV3 and YoloV3-tiny could achieve high mAP for corn plants from the V1 to V3 stage. The highest mAP@IoU0.5 was 98.07 at stage V2 (Table 2). However, plant canopy overlapping was a major issue for large plants, where part of the whole plant was usually considered as a single object. The mAP@IOU0.5 of the V4 stage only reached 63.13 for YoloV3-tiny, and 76.33 for YoloV3. Because of this, we suggest counting from top-view images for stages V1–V3. As for the V4 stage and after, side-view image detection will be our future study plan.

There was a trade-off in speed and precision. YoloV3-tiny could be an optimal solution for online counting. However, the precision needs to be improved to provide robust measurements for a Kalman filter estimator. We tried two different ways to improve the detection precision of YoloV3-tiny. The first method was to train the network to treat the four growth stages as four different classes, and the second method was to train the different stages as a sequence rather than combining them. These different training methods did improve mAP, especially for the V1 and V4 stages, and the results are shown in Figure 7.

Method 1 performed well at detecting the stages on its own. The mAP of four classes had significant improvement compared with the original YoloV3-tiny. Stage V4 reached the highest score in this method. In method 1, it was obvious that the four growth stages appeared like three different objects. The plants at the V2 and V3 growth stages were similar to each other. In method 2, we manually weighted the stages V1 and V4 in the detection network to improve the detection mAP. Because the detected mAP was lowest for stage V4, we first trained the network on stage V4 until its loss converged, then trained stages V2 and V3 sequentially, then finally trained the V1 stage. The detecting result reached the best scores for stages V1, V2, and V3, which were even over 99% for stages V2 and V3. Though the training time was about four times that of the original model, method 2 was still an optimal solution due to its high accuracies.

### 3.2. Track and Count

In this study, counting accuracy is defined as the formula (2). $Num_{GT}$ is the manual seedling count number, and $Num_{M}\;$is the seedling number counted through our pipeline. The reason not to divide $Num_{GT}$ by $Num_{M}$ directly is that the pipeline may count more or less than the ground truth. So we calculate the counting accuracy through the absolute error.
(2)$$\mathrm{Counting}\;\mathrm{accuracy}=\left(1-\frac{\left|Num_{GT}-Num_{M}\right|}{Num_{GT}}\right)\times 100\%\;$$

The count results (Figure 8) show that YoloV3-tiny can handle the detecting problem better with the customized training method, especially for stage V1. In the other growth stages, training method 2 was able to reach nearly the same count accuracy as YoloV3-tiny, but it counted four times faster than YoloV3. YoloV3-tiny’s average count speed was 47.1 frames per second, while YoloV3 only reached 10.7 frames per second. Aside from considering the aspect ratio changing in the state vector, we also adopted several strategies to improve the performance of the YoloV3-tiny-based counting method, including initializing the YoloV3-tiny detection model with high measurement uncertainty in a Kalman filter, and adding *max age* to recycle unmatched tracking IDs. We found that setting *max age* as 2 was best for this corn seedling counting task.

Figure 8 shows the counting result of the YoloV3 model and three method-trained YoloV3-tiny models. Our proposed method 1 and method 2 can improve significantly at the V1 stage and achieve nearly the same performance as YoloV3 at the V2 and V3 stages, while decreasing the counting accuracy standard deviation.

## 4. Discussion

Counting for corn plants at stages V2 and V3 achieved high accuracies of 98.66% and 98.06%, respectively. However, performance deteriorated when counting plants at stages V1 and V4. Detection precision for both stages V1 and V4 is lower than for the other two stages, but the main reason for this low counting accuracy is different. At stage V1, the YoloV3-tiny model can usually detect objects correctly but the Kalman filter fails to track them (Figure 9b). When detected bounding boxes are small, the values of *s* and *r* are small and change slightly along the time axis. In our discrete linear tracking system, the noise from motion is high as a result of the cart vibration and uneven road in the field, and this makes it difficult to track the small objects. Thus, the performance of counting is much lower than the detection. As for counting at stage V4, the counting accuracy decreases because of the overlapping and occlusions of the plant canopies. The similarity between V4, V2, and V3 will confuse the neural network. The YoloV3 model tends to detect a single corn plant as several objects (Figure 9c), and new tracking IDs are assigned to them. Thus, when they cross the finish line, the counting number is larger than the ground truth. That is the reason why counting accuracies are higher than detecting precisions when using YoloV3 and YoloV3-tiny original weight at stage V4.

Aside from the two aforementioned reasons, other random noise introduced by the detection network can also affect performance, such as missed detections (Figure 9d) and false detections (Figure 9c,e). However, these falsely detected bounding boxes generally have a very short life (≤2 frames). It is rare that a new tracking ID will be assigned to them, and even if one is assigned, it usually expires before they cross the finish line successfully.

To improve the performance of stand counting with corn plants at the V1 stage, the camera height may need to be adjusted to obtain detailed information for tracking. If one chooses a high-resolution camera to capture images, zooming into a suitable size on the region of interest will also work (but the network must also retrain based on the old weight). In addition, there are some advanced filters which have been developed to track small targets [52,53,54]. As for counting plants at stage V4 and later stages, the main issue is caused by the overlap of plant canopies in top-view images. In this case, detecting plant stems in side-view images may be a better option for stand counting.

## 5. Conclusions

In this paper, we present a pipeline for counting corn seedlings in the field at growth stages from V1 to V4. The pipeline combines a powerful high-speed one-stage detection network, YoloV3-tiny, with a Kalman filter. A customized training method has also been introduced to improve YoloV3-tiny’s detection accuracy. Experiments showed that this method can count corn plants at stages V2–V3 with accuracies over 98% under different outdoor lighting conditions, outperforming other CNN-based approaches for fruit counting [10,17]. Furthermore, the YoloV3-tiny-based model can achieve real-time counting. Thus, the pipeline is a cost-effective, high-speed, and real-time method. In addition, only one camera is needed for the system. We believe this pipeline can most likely be applied to count similar agricultural objects such as sorghum and soybean seedlings. There are still some improvements that can be made in this method, which is also our future study plan. For example, designing a height-changeable mechanical structure to suit different seedling heights could provide a big enough object (enough features) for the network to learn. A side-view of corn will be adopted to count the V4 stage and later stages, to overcome the overlapping problem.

## Figures and Tables

**Figure 1 sensors-21-00507-f001:**
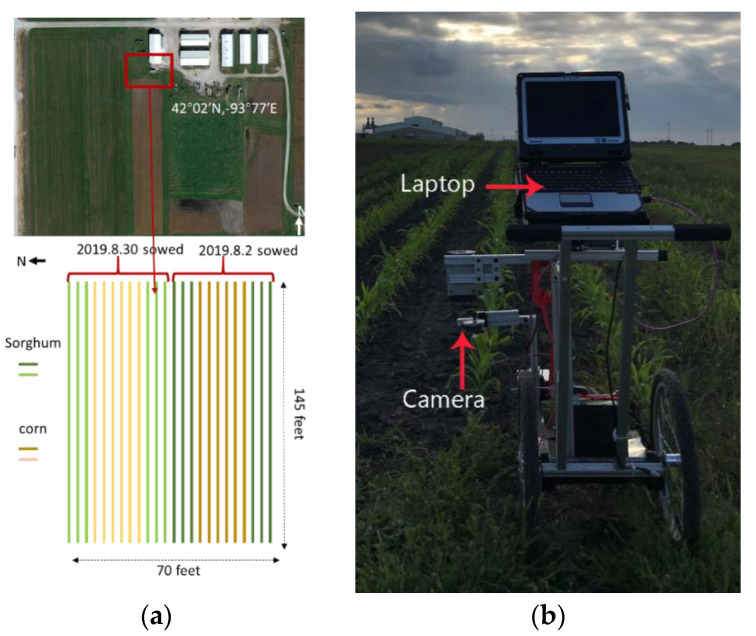
Diagram of the field layout and data acquisition system: (**a**) sketch map of sorghum and corn growth; (**b**) data collection cart.

**Figure 2 sensors-21-00507-f002:**
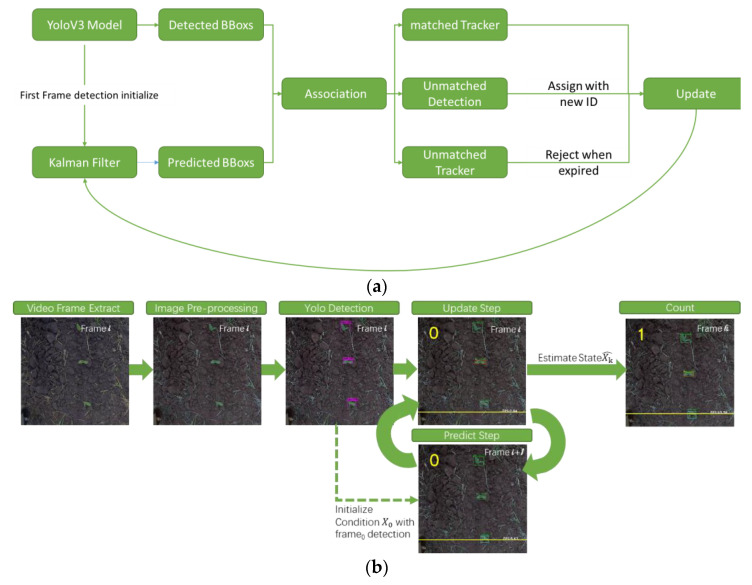
The counting pipeline (**a**) and the same shown with images (**b**).

**Figure 3 sensors-21-00507-f003:**
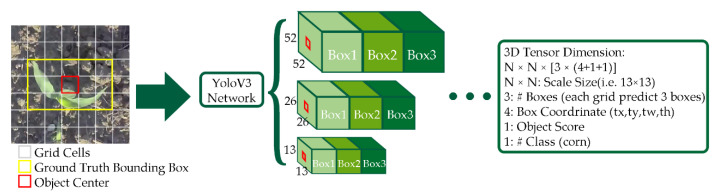
Prediction pipeline of a 416 × 416-pixel image at 3 scale sizes: 13 × 13 pixels, 26 × 26 pixels, and 52 × 52 pixels, with 3 bounding boxes for every cell. The grid cells are used for prediction. The red box and yellow box represent the object center and the ground truth, respectively.

**Figure 4 sensors-21-00507-f004:**
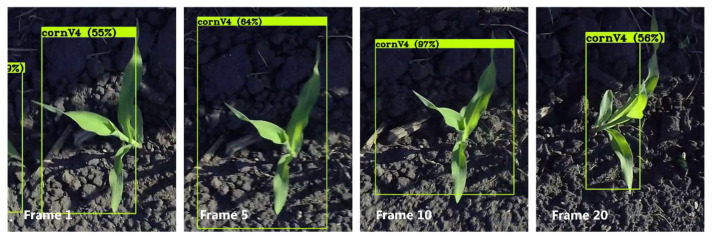
Example of the same object detected with different aspect ratio *r* along the movement axis.

**Figure 5 sensors-21-00507-f005:**
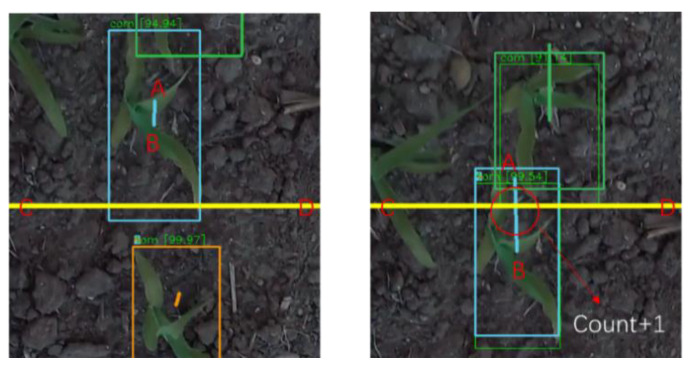
Demonstration of finish line for counting.

**Figure 6 sensors-21-00507-f006:**
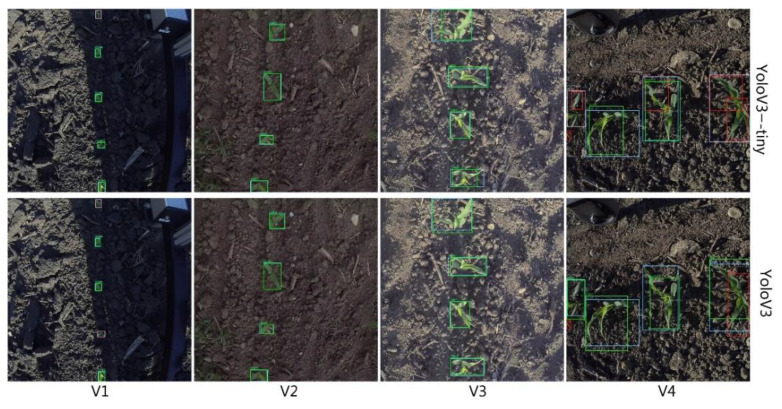
YoloV3-tiny and YoloV3 detection results for four growth stages. Green = TP: true positive (object detected and matches ground-truth), red = FP: false positive (object detected but does not match ground-truth), pink = FN: false negative (object not detected but present in the ground-truth), and blue = GD: ground truth (detected object’s ground-truth).

**Figure 7 sensors-21-00507-f007:**
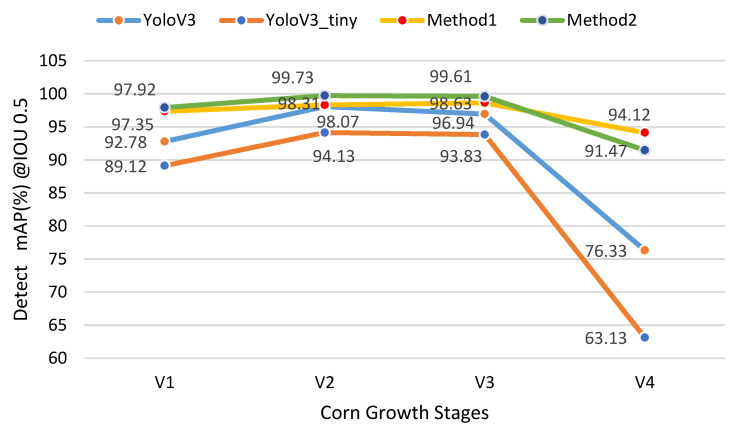
The mAP(%) of different training methods for YoloV3-tiny and YoloV3.

**Figure 8 sensors-21-00507-f008:**
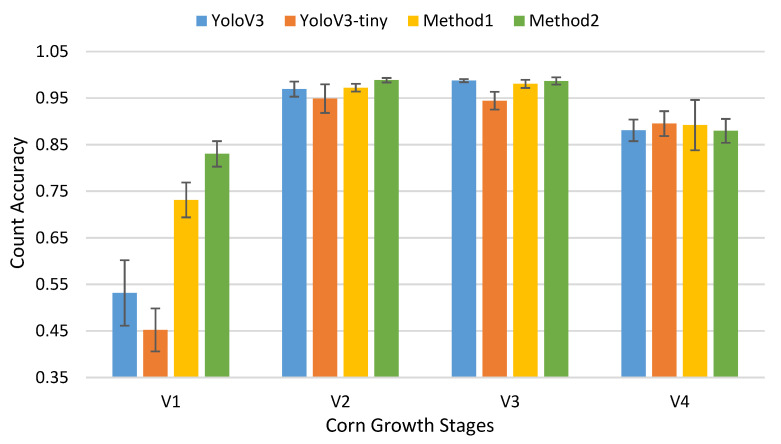
Count accuracy of YoloV3 and different training methods for YoloV3-tiny. The error bar represents the standard deviation of count accuracy.

**Figure 9 sensors-21-00507-f009:**
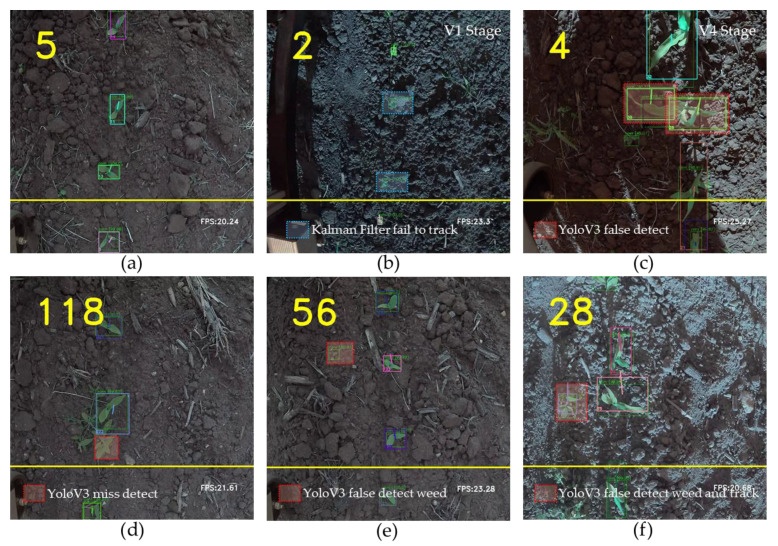
Summary of six different detection and tracking scenarios: (**a**) correctly detect and correctly track; (**b**) YoloV3 detects correctly but Kalman Filter fails to track; (**c**) detect one object as three objects and track them; (**d**) missed detection; (**e**) false detection of weed as corn; (**f**) false detection of weed as corn and track.

**Table 1 sensors-21-00507-t001:** Summary of platforms and sensing technology for stand counting in different crops.

Camera/ Platform	Study Object	Camera Height	Resolution	Detect /Classify	Count Performance	Reference
Hyperspectral /UAV	Cotton stand	50 m	1088 × 2048	Segment according NDVI	Accuracy = 98% MAPE = 9%	Feng et al. (2019) [11]
RGB/UAV	Potato emergence	30 m	20 megapixel	Edge detection	R^2^ = 0.96	Li et al. (2019) [23]
RGB/UAV	Rice seedling	20 m	5427 × 3648	Simplified VGG-16	Accuracy > 93% R^2^ = 0.94	Wu et al., (2019) [10]
RGB/UAV	Winter wheat	3~7 m	6024 × 4024	SVM	R^2^ = 0.73~0.91 RMSE = 21.66~80.48	Jin et al. (2017) [42]
RGB/UAV	Rapeseed seeding stand	20 m	7360 × 4912	Shape feature	R^2^ = 0.845 and 0.867 MAE = 9.79% and 5.11%	Zhao et al. (2018) [12]
RGB/UAS	Cotton seeding	15~20 m	4608 × 3456	Supervised Maximum Likelihood Classifier	Average Accuracy = 88.6%	Chen et al. (2018) [8]
RGB/UAS	Corn early-season stand	10 m	6000 × 4000	Supervised Learning	Classification Accuracy = 0.96	Varela et al. [13]
Multispectral /UAV	Corn stand	50 feet	0.05 m ground spatial	“Search-Hands” Method	Accuracy > 99%	Rawla et al. (2018) [41]
RGB/Ground level (handheld, cart, and tractor)	Cotton seeding	0.5 m	1920 × 1080	Faster RCNN Model	Accuracy = 93% R^2^ = 0.98	Jiang et al. (2019) [17]
RGB/moving cart	Wheat density estimation	1.5 m	4608 × 3072	Gamma Count Model	R^2^ = 0.83 RMSE = 48.27	Liu et al. (2019) [43]
RGB/Hand-held camera	On-ear corn kernels	/	1024 × 768	Semi-Supervised Deep Learning	MAE = 44.91 RMSE = 65.92	Khaki et al. (2020) [30]

**Table 2 sensors-21-00507-t002:** Evaluation of YoloV3 and YoloV3-tiny classification on four growth stages.

mAP(%)@IoU0.5	V1	V2	V3	V4	Average FPS
YoloV3	92.78	98.07	96.94	76.33	12.21
YoloV3-tiny	89.12	94.13	93.83	63.13	76.27

## Data Availability

Data sharing not applicable.

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
