# Peer review of "A Convolutional Neural Network-Based Method for Corn Stand Counting in the Field"

_sensors, 2021, doi:10.3390/s21020507_

Round 1

Reviewer 1 Report

Stand counting is an important task in breeding and crop management, providing farmers and breeders with information about the germination rate and population density of crops. The traditional method for stand counting usually samples the corn population manually with quadrats, which is time-consuming, laborious, and cost-intensive. This study presents a machine learning-based image processing pipeline for detection and counting early-season corn seedlings in the field, which could be integrated into an unmanned robotic system for plant growth monitoring and phenotyping. The pipeline combines YoloV3-tiny detection network with Kalman Filter, achieving an accuracy over 98% at growth stage V2 and V3 with an average frame rate of 47 frames per second. The results demonstrate that the method is accurate and reliable for stand counting.  

General comments:

  1. L53-L57: The relationship between the research in Table 1 and the two methods mentioned above is not reflected in the table.
  2. L76-L77: The quotation in Reference 15 is incorrect, Faster R-CNN achieves target detection
  3. L98: How to ensure that the speed of pushing the trolley is 1m/s in the research?
  4. L100-L106: The VE, and VT defined in this paragraph are not quoted in the following text.
  5. L114-L115: There are 80 images in the data set that have not been allocated.
  6. L173: Please describe that's' refers to the ratio between the rectangular box and what.
  7. L187: Is the detection accuracy of YoloV3 sufficient to satisfy the counting requirement?
  8. L193-L195: What is the relationship between tracking ID and counting ID?
  9. L216: According to the description in the text, ‘YoloV3-tiny’ could be an optimal solution for online counting than ‘YoloV3’.
  10. What do the symbols on each bar chart in Figure 8 represent?
  11. The confusing references to ‘detection accuracy’, ‘detection precision’, and ‘counting accuracy’ in the article resulted in the inability to distinguish between analysis and detection conditions or counting conditions. In addition, there is no definition of ‘counting accuracy’ in this article, so it is difficult to understand the ‘decrease in counting accuracy’ described in line 257.

Specific comments:

  1. L74: Change ‘has’ into ‘had’.
  2. L109: Please give a rough description of Fig. 1a
  3. Figure 3: The font in Figure 3 is not clear, such as ‘N×N’, ‘X×X’, or ‘N×X’.
  4. L210: Change ‘tiny’ into ‘V3-tiny’.
  5. L213: Add ‘stage’ before ‘only’.
  6. Table2: Change ‘Table.2.’ into ‘Table 2.’
  7. 7. Please describe in the text before inserting Table 2.

Reviewer 2 Report

The manuscript presents a method based on YoloV3 network and Kalman Filter to count corn seedlings during all growth stages.

It is very well written, and only a few minor changes are required before being accepted.

Comments:

A) The text is well written, and the authors should consider breaking some paragraphs in two, to make it look even better. Please take a loog at the following paragraphs:

Paragraph starting on line 52 and ending in line 68;

Paragraph starting on line 69 and ending in line 83;

Paragraph starting on line 120 and ending in line 137;

Paragraph starting on line 140 and ending in line 145;

B) You mention that the noise introduced by the cart vibration and uneven road was high, making it difficult to track small plants.

Do you believe that a small AUV would solve this problem (since they are easily affected by wind gusts), or a large and much more stable AUV would be required? Please comment, and if appropriate, plase include in the manuscript. 

B) If possible, discuss the cost trade-off between using a tractor (which will probably present the same counting response of your manual cart, with a lower sMAP) and a large and stable UAUV. It would be very intreresting to see this discussion.

C) How important is the speed? Would the system benefit if another version of the Intel Xeon CPU was used? For example, the  Xeon E5- 2670 (that has much more cores, althoug it  runs at a slightly low clock frequency) instead of the E5-2637?

To be revised:

Abstract - "...at growth stage V2 and V3". 

Please explain the meaning of the V2 and V3 stages in the abstract. There is no limitation to the abstract lenght, so the manuscript will benefit from this. 

Line 114 - A copy&paste mess here...? Please rewrite.

"The labeled images were split into train, validation 722:142, with a proportion of."

Line 117 - There is an email link in the CPU's name. Please revise it.

Line 177 - "...tracking state model for normal bounding boxes [15,47], which takes the aspect ratio r as a constant,"

Please use italic for the variable "r"; note that it appears in other senteces,

Line 200 - "...had high mAP(mean Average Precision)"

  • Space is missing:  mAP(mean Average Precision)
  • Please define mAP
  • "mAP@IoU 0.5". lease define and explain.

Reviewer 3 Report

The paper proposes an approach based on a convolutional neural network for corn stand counting in the field. The paper is interesting and easy to read. The video attachment is very useful to the reader to better understand the proposed application. However, the following points need to be improved before publication.

1) The pros and cons of the proposed approach should be better highlighted with respect to the present literature.

2) The computational effort of the method shold be better described. It would be interesting to mention the size of data acquired and processed for the esimation of corn number for fized distances (How many GB for 100 feet?)

3) The data acquisition system and the sensors used in the application should be better described to help the reader in the replication of the experiments.

4) In the literature review, other sensing technologies (such as Lidar, Optrx crop sensors) should be mentioned in the context of precision farming and plant counting. I suggest the following articles:

Pantano, M., Kamps, T., Pizzocaro, S., Pantano, G., Corno, M., Savaresi, S. (2020). Methodology for Plant Specific Cultivation through a Plant Identification pipeline. In 2020 IEEE International Workshop on Metrology for Agriculture and Forestry (MetroAgriFor) (pp. 298-302).

Ristorto, G., Gallo, R., Gasparetto, A., Scalera, L., Vidoni, R., & Mazzetto, F. (2017). A mobile laboratory for orchard health status monitoring in precision farming. Chemical Engineering Transactions, Volume 58, 2017, Pages 661-666.

Round 2

Reviewer 3 Report

The paper was improved as suggested and it can be accepted for publication.